# Effect of Personal and Contextual Factors of Regulation on Academic Achievement during Adolescence: The Role of Gender and Age

**DOI:** 10.3390/ijerph18178944

**Published:** 2021-08-25

**Authors:** Jesús de la Fuente, Erika Andrea Malpica-Chavarria, Angélica Garzón-Umerenkova, Mónica Pachón-Basallo

**Affiliations:** 1School of Education and Psychology, University of Navarra, 31009 Pamplona, Spain; mpachonbasa@alumni.unav.es; 2School of Psychology, University of Almería, 04120 Almería, Spain; 3School of Psychology, Fundación Universitaria Konrad Lorenz, Bogotá 110231, Colombia; erikaa.malpicac@konradlorenz.edu.co (E.A.M.-C.); agarzonu@gmail.com (A.G.-U.)

**Keywords:** self-regulation, self-efficacy, procrastination, educational level of family, gender, adolescence, academic achievement

## Abstract

This investigation aimed to analyze the predictive differential value of personal (self-regulation, self-efficacy, procrastination) and contextual characteristics (parents’ socio-educational level), regarding academic achievement, among Colombian adolescents. A total of 430 students (from 11 to 18 years old) from both genders filled out validated self-reports and informed their academic achievement. We performed an ex-post-facto design, simple regression analyses, structural equations predictions analyses (SEM), and variance analyses (ANOVAs). The results showed that self-regulation is the most potent personal variable predictive of procrastination and achievement, positively associated with self-efficacy; additionally, the parents’ educational level was also a predictor, although to a lesser level. The female group and the elderly group negatively predicted academic achievement, behaving as modulatory variables of the above results.

## 1. Introduction

Previous investigations have shown that academic achievement among adolescents depends on personal factors like age and gender and familiar factors as upbringing, all related to their adequate development and emotional well-being [1,2,3,4,5].

Academic achievement refers to the grades achieved by a student at school. It represents the achieved learning level associated with cognitive, social, motivational, and cultural factors [3,6,7,8,9,10] and can be expressed in quantitative or qualitative values, in accordance to the adopted system by each educational institution [3,6,7,8,9,10,11]. Habitually, this condition is measured with standardized tests [4] that, through numerical grading or value judgments, reflect a student’s capacity, that depends on personal, familial, social, and institutional aspects [10]. Understanding the factors associated with academic achievement in adolescents is necessary to comprehend the individual differences integrally to promote academic success and reduce academic desertion [5,6,12].

The present investigation aims to predict academic achievement in high-school Colombian adolescents considering personal variables such as age, gender, self-regulation, procrastination or academic self-efficacy, and familiar variables such as parents’ educational level.

### 1.1. Academic Achievement during Adolescence: Changes According to Gender and Age

Li et al. [10] mention a reduction in motivation in the transition from primary school to high school. This condition affects academic achievement and increases stress due to the changes in the institutional, curricular, and requirements environment [8].

Some investigations have suggested *gender* as one of the conditions that influence academic achievement, as evidence supports differences between men and women in essential sciences learning. For example, the Organization for Economic Co-operation and Development (OECD) report [13] concerning the achievement on the PISA test in 2015 pointed out that men obtained a superior achievement compared to women in the total test score. In the case of Colombia, men had an outstanding achievement in math, and women performed better in areas related to language [14,15]. However, Correa (2016) [16] highlights that the TIMSS 2007 tests (Trends in International Mathematics and Science Study) indicate that gender is not a determinant factor on academic achievement.

On the other hand, it has been affirmed that in low-income contexts, beliefs of higher success and self-efficacy expected of men are legitimated, which can be explained by the bias of answers when filling out the instruments [17,18]. The latter, in turn, can be reflected in male self-confidence regarding math achievement, so the formation of negative beliefs concerning women’s achievement is problematic to the point that it reduces their motivation [9,16].

This way, the PISA 2018 results report showed a tendency of women to express increased fear of failure, associated with acquiring worse learning results in math. According to this report, self-efficacy and achievement are closely related in men and women [19]. Nevertheless, again, the results are contradictory. For example, in a study involving chemistry students, there were no significant differences in the gender associated with self-efficacy and academic achievement [20].

Regarding learning self-regulation, it has been shown that women tend to use strategies such as supervision, goal planning, time management, constant monitoring of the already learned and weakness identification, aiming to reinforce knowledge [17], these conditions reflect a higher sense of responsibility than some men [5,18,21].

Concerning age, it has been evidenced that adolescents feel overwhelmed by schoolwork; they perceive less support and satisfaction with their peers and faculty, a situation that increases academic stress [22,23]. On the other hand, Ruiz and cols. [23] indicate that as age and academic courses progress, there is an increase towards goals of social reinforcement. However, the investigation by Delgado et al. [24] suggests that the differences in learning goals are minor among advanced or introductory courses.

### 1.2. Students’ Factors of Regulation: Self-Regulation, Procrastination and Academic Self-Efficacy

According to the OECD report [19], high school students who routinely fail in their academic tasks usually present focus and emotional management problems that lead them to procrastinate. In addition, it has been demonstrated that a vast number of adolescents have issues meeting deadlines, do not prepare for midterms and readings, and avoid their regular school duties [25,26].

Self- vs. externally-regulated theory, SRL vs. ERL [27] has proposed that improving students’ achievement, self-regulation, and regulatory context is essential. At the *individual level*, self-regulation (SR) is a presage variable or personal characteristic, and general meta-ability, that people have to respond to vital demands with adequate control of their emotional reactions. It has been defined as an intrapersonal (individual) variable that allows people to manage their decisions, making it possible for them to plan, exercise control over such decisions, and evaluate their effects. This SRL vs ERL theory [27] considers that self-regulation (SR) is a prior personal characteristic that facilitates self-regulated learning (SRL), as process, in the academic achievement; this way, self-regulation (SR) is a constructive process that allows students to establish goals to monitor, regulate, and control their behavior [28]. Those actions make them persevere in the assigned tasks, construct, and regulate their behavior, even in situations that do not reinforce behavioral self-regulation [29].

That way, those who possess self-regulation and control skills choose adequate behavioral adjustment strategies, self-control the development of their working schedule looking to respond appropriately to their tasks to improve their performance [8,30]. The self-regulation general variable is present in the promotion of motivational–affective strategies [28,29]; the type of assumed learning scope [31,32,33]; achievement emotions [34,35]; coping strategies [36]; academic confidence and procrastination [37]. Contrary to self-regulation, when a dys-regulatory behavior as procrastination is present [26,27], people decide to postpone actions regardless of their adverse effects, increasing the perceived stress level, affecting important adjustment behaviors. This situation can even affect mental health [38,39].

In a school context, academic procrastination combines affective, cognitive, and behavioral mechanisms in response to the perceived psychological pressure concerning the achievement of school activities considered aversive. The latter is why those activities are not done on time or with quality, affecting academic achievement [40,41,42,43]. In the adolescent population, the delay in schoolwork has been associated with low self-esteem, poor self-regulation, low self-efficacy, and other factors such as laziness, difficulty in decision making, exposure to risks, and anxiety [25,26,44,45,46,47].

Finally, several investigations point out that academic achievement is related to students’ beliefs about their personal and social capacities, especially those related to the general self-efficacy and the learning faculties, capabilities that consolidate throughout childhood and adolescence [9,10,38,48,49].

The Bandura self-efficacy theory refers to the conviction that people have to successfully fulfill a task following the self-imposed goals [50]. Schöber et al. [48] propose that self-efficacy in the academic context implies issuing individual judgments to organize and execute courses of action to develop assigned and chosen educational activities and define the effort level and persistence.

Adolescents value their achievement from the perceived capacity to achieve personal academic goals; if they believe that the task surpasses their skill, the risk of failure increases. The inability to overcome difficulties is a situation that affects academic self-efficacy [9,10,12,20], which in turn is considered as the cause and effect of academic achievement and is also a predictor of academic success, especially when the adolescents have a policy context of peers with whom they compare to [12,38,51].

### 1.3. Family Context Factors of External Regulation: The Socio-Educational Level of Parents

At the *contextual level*, the SR vs ER Theory model has also defined the external-regulatory/non-regulatory/dys-regulatory role of the context. Recently, it has been shown that the low/medium/high level in contextual regulation are predictors to the external-regulatory/non-regulatory/dys-regulatory personal behavior [52].

The upbringing and the academic follow-up that youngsters receive facilitate their learning and discipline in educational contexts [3,53]. Therefore, parents stimulate their children to achieve important goals, interest in academic tasks, and adaptation to school structure and communication with faculty [20,54,55]. The familiar and school context in which the student develops is foundational to the early building of adaptive motivational and affective strategies to their academic confidence and procrastination management [29,42,56,57,58,59,60,61,62].

Among the main familiar characteristics associated with academic achievement are the educational and socio-economic levels. The assignment of economic, environmental, and intellectual resources from parents to children are related to positive perceptions of care, autonomy reinforcement, academic competence, support through advice, help with tasks, and information availability [2,7,11].

However, the investigation by Leal-López et al. [63] in 33 countries with adolescents between 11 and 15 years old concluded that living with families of high material affluence represents a risk factor for increased alcohol consumption. In fact, among adolescents with high wealth levels, factors such as excessive pressure for goal achievement and their parents’ emotional and physical distance make them vulnerable to substance abuse, depression, and anxiety than those of medium or low-income levels [64].

On the other hand, poverty, lack of resources, violence, and the parents’ low academic level have been negatively associated with academic achievement, access to higher education, and future income.

However, authors such as Luthar and Latendresse [64] mention that the stereotype vision that the high or low psychosocial risks among adolescents are directly associated with being poor or rich is erroneous, as the comparative studies show more similarities than differences among adjustment patterns and socialization processes. Up to a point, and independently of income level, the parents who share academic activities with their children positively affect academic achievement. The latter is a subject that needs further research.

It is worth noting that parents with high educational level increase their children’s cognitive development and improve academic achievement [1,4,14,16,17]. The estimation or capability to perform an effective upbringing influences the integral development of kids and adolescents and promotes self-regulation [4,21,28,65].

### 1.4. Objectives and Hypothesis

From the problem mentioned above, the *objective* in this investigation was to test a predictive model of academic achievement in adolescence that allows us to assess: (1) the predictive value of students’ factors about gender and age; (2) the predictive role of self-regulation and variables associated differentially to it, such as self-efficacy and procrastination; (3) the predictive weight of the families socio-academic level—all the above in a population of Colombian students. As a consequence of the objectives, the investigation hypothesis was of two types.

### 1.5. Predictive Lineal Hypothesis

(1) Previous research has shown that gender and age are relevant variables that affect academic achievement in adolescence [5]. Therefore, it is expected that both variables appear to modulate achievement. Age will predict achievement negatively, while gender (M = 1, F = 2) will positively predict it.

(2) Previous research has shown a negative prediction between SR and procrastination (Garzón et al., 2018) [66]. Therefore, it is expected that the SR will be negatively predict procrastination and positively academic achievement, together with self-efficacy expectations. Finally, socio-educational context (parent’s academic level and work type) will positively predict achievement.

### 1.6. Inferential or Non-Lineal Hypothesis

(3) The students’ self-regulation (SR) level (low/medium/high) will predict the self-efficacy, procrastination (inversely), and achievement levels. Complementary, the socio-educational context will determine the achievement level, although less strongly than SR.

## 2. Method

### 2.1. Participants

The study included 430 high school students (sixth to eleventh grade) ranging from 11 to 18 years old (M = 14.50, SD = 1.9). They belonged to a public school in Bogotá, 193 men (44.9%) and 237 women (55.1%). A convenience sample was performed, guaranteeing sample heterogeneity. The participants and their legal guardians expressed their will to take part in the study, therefore signing an informed consent previously approved by the Bioethical Committee of the Konrad Lorenz University. Likewise, the school directives were informed and agreed to the study.

### 2.2. Instruments

*Self-regulation*. The study used the second Spanish version of the Self-regulation Questionnaire (SSSRQ). This version adapted by Pichardo, et al. (2018) [67] has 17 items that measure self-regulation of general behavior and was adapted to adolescents from ages 12 to 17. The answer options are on a 5-point Likert scale, as follows, 1 (Strongly Disagree), 2 (Disagree), 3 (Uncertain or Unsure), 4 (Agree), 5 (Strongly Agree). Results indicate that the proportion of variance explained for the factorial model of the SSSRQ was 86% for all the items. The factors also explained adequate percentages of the variance: goal-setting (90%), learning from mistakes (88%), perseverance (84%), and decision-making (78%). The four factors obtained an internal consistency of 0.84 to 0.95. The Rasch analyses showed that all items in the four subscales present a good fit to the model. The adjustment indices of the self-regulation questionnaire are higher than 0.90, squared 641.209, CFI = 0.992, TLI = 0.994 and RMS = 0.075; conditions that indicate that the version in Spanish validated for the Colombian population meets the conditions of construct validity.

*Procrastination*. The Procrastination Assessment Scale-Student (PASS) by Solomon and Rothblum (1984) [68] in its Spanish version and adapted to the Colombian population by Garzón and Gil (2017) [69] was used. The scale has 44 items divided into two sections; the first has 18 items and assesses procrastination frequency and the related anxiety degree; the second goes from item 19 to 44 and inquires the cognitive-behavioral reasons and motives to procrastinate. As in the original, the adapted version measures the appearance level and the use of motives to postpone and is grouped in 5 sub-scales as follows: search for excitement, lack of energy and self-control, perfectionism, evaluation anxiety, and low assertiveness and confidence. The answering options are a Likert scale, with five different possible values as follows: 1 (never/does not reflect my motives whatsoever), 2 (rarely/reflects my motives a little) 3, (sometimes/reflects my motives to a point), 4 (almost always/reflects my motives greatly), and 5 (always/reflects my motives perfectly). In the analysis of the results in the present study the following scores were obtained: (a) intensity level (procrastination frequency), as the sum of Questions 1, 2, 4, 5, 7, 8, 10, 11, 13, 14, 16, and 17; (b) grouped scores of each one of the three reasons to procrastinate (Questions 19 through 44); and (c) total procrastination, as the sum of the previous two. For the test validation in Colombia, a linguistic adjustment was made, and adequate reliability values were obtained (Cronbach’s alpha of 0.71–0.82); also, discriminant validity evidence was obtained for the procrastination frequency in function of time management and academic achievement measures (Garzón and Gil, 2017).

*Self-efficacy*. The Perceived Self-efficacy on Academic Situations Scale constructed by Palenzuela (1983) and adapted to Colombian population by Garzón, Gil and de Besa (2021) [70] was used. The scale is composed of 10 items that measure the students’ self-efficacy expectations on academic contexts. The answering options are on a 4-point Likert scale as follows: 1 (never), 2 (sometimes), 3 (often), and 4 (always). The initial version has an adequate functioning level with a Cronbach’s alpha of .91. In different adaptations of the test in Latin American countries, the alpha values are between 0.89 and 0.91 [71]. For the Colombian population, adequate reliability values for people (0.89) and for the measure (0.99) were found, as well as good fit to the Rasch model, and good Likert scale functioning [70]. The fit indices indicate that the instrument has a high level of construct validity reflected in the main psychometric indicators (χ² = 542.41; gl = 35; RMSEA = 0.06; RMR = 0.03; GFI = 0.97; AGFI = 0.95; NFI = 0.95; TLI = 0.94 and CFI = 0.96).

*Academic Achievement.* Students reported their average grade for the previous year (range: 1–10 points). Each year the average grade of the student is calculated, from all the averages obtained in all the subjects, to put in their academic record. Although these data were self-reported by each student, it is reliable as it is the grade from the school record. We assume the grading as the indicator for academic achievement.

*Socio-demographic factors*. A socio-demographic questionnaire that collects information related to personal and socio-demographic conditions (grade, age, gender, academic achievement, and parent’s academic level) were used. The calculation of the family’s value of the socio-educational context is the average of the educational level of the mother and the father (range: 1–6 points) and of work type (range: 1–15 points).

### 2.3. Procedure

In the first place, the researchers obtained authorization from the educational institution directorate. For that, a commitment to share with the institution the study results at the end of the research period was established. There were no economic incentives for the students or parents to take part in the study. Secondly, through the schools counseling department, consents for the underage students, parents, or legal guardians were obtained. It was clarified that the information would be employed for academic purposes and was not to affect the participant’s grades. Thirdly, the information was collected online in two months using Microsoft Forms on the TEAMS platform, which was a resource made available for the students by Bogota’s Education Secretariat. Fourthly, during the data collection, a follow-up was carried out to remind course directors and parents of the availability of the questionnaires. Finally, with the obtained data, we proceed to the statistical analyses.

In alignment with the Colombian Deontological Code (Bill 1090 of 2006) and the 8430 Mandate of 1993, a guarantee of anonymity and confidentiality of the collected information was given to the sample’s total [72].

### 2.4. Data Analyses

The use of an ex post facto transversal design [73,74] allowed three types of analyses. First, we explored the quality of the data by examining the existence of outliers and missing cases. The method used for detecting univariate outliers consisted of calculating the typical scores of each variable and considering cases with Z scores outside the ±3 range to be potentially atypical cases [73]. On the other hand, we used the Mahalanobis distance (D²) to detect atypical combinations of variables (atypical multivariate cases), a statistical measure of the multidimensional distance of an individual, concerning the centroid or mean of the observations given [74]. This procedure detects significant distances from the typical combinations or centroids of a set of variables. The literature suggests removing univariate and multivariate outliers or recodifying them to the nearest extreme score [75]. The procedure was carried out using SPSS (26, IBM, New York, NY, USA), which provides a specific routine for missing values analysis that determines the magnitude of missing values and whether they are presented systematically or randomly.

We also evaluated assumptions related to sample size, independence of errors, univariate and multivariate normality, linearity, multicollinearity, recursion, and interval measurement level, all presenting acceptable levels of reliability. Regarding the sample size, the inclusion of 10–20 cases per parameter is recommended, and at least 200 observations [76]. Independence of errors refers to the fact that the error term of each endogenous variable must not be correlated with other variables. To determine univariate normality, we examined the distribution of each observed variable and its indices of asymmetry and kurtosis. Asymmetry values greater than three and kurtosis greater than 10 suggest that the data should be transformed [76]. On the other hand, values less than 70 on the Mardia multivariate index indicate that distance from the multivariate normality is not a critical deterrent to this analysis [77,78]. Finally, although one of the assumptions is interval measurement, in some cases, variables measured at a nominal or ordinal level are used, as long as the distribution of scores, particularly of the dependent variables, is not markedly asymmetric [73].

The multicollinearity assumption was analyzed by examining bivariate correlations between variables since a correlation of 0.85 or higher would indicate difficulty fulfilling this assumption. Furthermore, the model should be recursive, so causal influences must be one-directional and without retroactive effects. Finally, it is recommended that the measuring instruments show at least moderate reliability properties. This aspect was also fulfilled (see instruments section).

As a preliminary analysis, we checked for normal sample distribution using the Kolmogorov–Smirnoff test for dependent variables. We also used the Hoelter Index to determine sample size adequacy [73]. In addition, we conducted analyses of linearity and atypical values, missing and influential cases, as well as critical values of multivariate normality; recommended values for the multivariate index of kurtosis, or Mardia coefficient, are less than 70 [78].

For Hypotheses 1 and 2, Pearson bivariate correlations were carried out. Complementary, we used predictive analyses of structural equations or SEM models. For this purpose, we followed the recommendations of Hu and Bentler [79], where a model shows a good fit to the observed data when the ratio of the Chi-square statistic to its degrees of freedom is less than five, the values of RMSEA and SRMR are <0.08, and NNFI (non-normed fit index), IFI and CFI are >0.95 [80]. Furthermore, if sample sizes are equal to or less than 250 participants, Hu and Bentler [78] suggest using only the CFI and SRMR fit indices (this was not our case). As an estimation method, the robust maximum likelihood method was used, which allows the use of polychoric correlations; their use is more suitable in variables with the preceding characteristics of high normality indices and multivariate kurtosis, and an ordinal nature [81]. We also examined the reliability of the model dimensions for the total and each of the proposed factor structures by calculating Cronbach’s alpha. The software programs used to perform these analyses were SPSS 26 [82] for the reliability analyses and AMOS v. 23 [83] for the confirmatory factor analyses and SEM model.

For Hypothesis 3, the socio-educative context variable, the cluster analyzes were performed to delimit the low-medium-high groups, both in the SR variable and in the family’s socio-educational context (average of the educational level of the mother and the father and of work type). Subsequently, simple and multiple ANOVAs were carried out to predict the effect of these levels on the defined dependent variables. The software programs used to perform these analyses were SPSS 26 (IBM, New York, NY, USA) [82].

## 3. Results

### 3.1. Previous Analyses

The results were adequate for verifying assumptions of normality in the different indicators analyzed. See Table 1.

### 3.2. Linear Association

Age was negatively correlated with academic achievement (*r* = −0.022, *p* < 0.05). However, self-regulation (SR) was negatively and significantly correlated with procrastination (*r* = −0.484, *p* < 0.001), and positively with self-efficacy (*r* = 0.559, *p* < 0.001) and with academic achievement *r* = 0.494, *p* < 0.001). Procrastination was negatively correlated with self-efficacy (*r* = −0.318, *p* < 0.01), with the father’s education level (*r* = −0.108, *p* < 0.05) and with academic achievement (*r* = −0.427, *p* < 0.001).

Additionally, self-efficacy was positively correlated with the mother’s and father’s education (*r* = 0.168, *p* < 0.001; *r* = −0.166, *p* < 0.001, respectively) and as expected, with academic achievement (*r* = 0.377, *p* < 0.001).

Other considerations show that the mother’s and father’s education correlate positively and significantly among each other (*r* = 0.40, *p* < 0.001), likewise, the parent’s education level correlated with academic achievement (mother: *r* = 0.201, *p* < 0.001; father *r* = 0.191, *p* < 0.001). It is important to point out that neither gender nor job correlated significantly with any other analyzed variables. See Table 2.

### 3.3. Linear Structural Prediction

We tested three structural models. *Model 1* (four factors) evaluated the predictive effect of personal variables (self-regulation, procrastination, self-efficacy) on achievement. *Model 2* (seven factors) evaluated the predictive effect of personal variables (self-regulation, procrastination, self-efficacy) and contextual (educational level of the mother and the father and of work type) factors on achievement. *Model 3* (nine factors) evaluated the predictive effect of personal variables (self-regulation, procrastination, self-efficacy) and contextual (education of mother), with two moderator factors (age and gender) on achievement. The third model showed the best statistical standards and offered the best indexes. See Table 3.

The results showed that the most predictive variables of achievement are self-regulation (SR), and self-efficacy (SEF). Likewise, there was an immediate negative effect of self-regulation on procrastination, but not of self-efficacy. In addition, gender (in favor of females) and the mother’s academic level had a direct positive effect on achievement. Meanwhile, the father’s educational level and work type appeared to have an indirect positive effect on achievement. Finally, although with less statistical relevance, there was a positive association between the father’s work type with the student’s self-regulation level. See Table 4 and Figure 1.

### 3.4. Inferentials: Self-Regulation Level and Socio-Educational Context Effects

Previous analyses, like the Box test of equality of multiple variance–covariance matrices and the Levene test for equality of error variances, did not show significant differences among groups.

The ANOVA and MANOVA 3 × 3 results (SR level × socio-educational context level) showed different significative effects. On one side, there was an effect of self-regulation (SR) level on the different analyzed variables (procrastination, self-efficacy, achievement); like so, the greater the SR level had a higher self-efficacy level, a lower procrastination level, and a higher achievement level. Complementary, a higher parent’s education had a higher achievement level. See direct results and specific effects in Table 5 and Figure 2.

## 4. Discussion

The general objective of the present study focused on the analysis of the predictive factors of academic achievement in a sample of high school Colombian students and to validate the self-regulation and external regulation effect on self-efficacy, procrastination, and achievement behaviors.

### 4.1. Hypothesis Discussion

Regarding Hypothesis 1, we proved that gender is associated with and predicts academic achievement negatively for women. Our results are consistent with previous studies that indicated that adolescent women usually have lower educational achievements when compared to men in areas like science and math [13,15]. Furthermore, the obtained results are closely related to familiar, socio-economic, and cultural conditions, as highlighted in other studies in the Colombian population [15,16]. However, it is essential to note that there are no conclusive studies in this regard. Authors such as Santos et al. (2020) [5] declare that women obtain better academic results because of the learning strategies and study habits they usually employ.

On the other hand, we found that adolescents between 15 and 18 years old performed worse on academic achievement. This condition can be explained considering that in this stage of the vital cycle, the adolescent faces a de-regulatory and critical period, a conflictive school stage due to puberty, which involves gaining control and independence. As a result, adolescents might feel anonymous and disconnected, despite the effort put during these transitions as they experience new inter-personal contexts, competitive love relationships, and the development of social and cognitive skills [84,85,86,87,88,89].

In Colombia, 17% of the population are children and adolescents. Besides the inherent matters of adolescence, a part of this population is exposed to a social and poverty context or to different types of violence [90] conditions that increase vulnerability, the acquisition of de-regulatory conduct, and poor academic achievement. The latter is especially true for those who live in chaotic homes compared to children and adolescents with protective environments that promote a satisfactory academic achievement [28,90].

As predicted on Hypothesis 2, self-regulation is associated and positively predicts self-efficacy and achievement, contrary to procrastination. These results have been widely evidenced in college students [37,61,91,92,93,94] and are also present in high school students. For that reason, due to the recent rise in virtual education for primary and high school students, it is essential to guide students to plan adequate knowledge acquisition goals and strategies of self-learning [95].

Regarding the familiar context, the family education level considered an external regulatory factor, also appeared as a predictive factor for achievement. However, it is not predictive of the other analyzed factors. These results are also consistent with previous evidence, which affirms that family functions as a determinant element for the favorable achievement of academic results, highlighting maternal education as a predictor of academic achievement, especially in kids and adolescents [96].

Nerona (2020) [97] showed in a recent study, those parents that have more education years take shared decisions concerning their kids’ education; maternal support is notably associated with the motivation to fulfill adolescents’ academic goals. Previous evidence indicated the existence of an association between the mothers’ education level and academic achievement. The mothers’ high education level seems to favor good upbringing practices that fit their children’s needs, more cognitive stimulation, and greater involvement in school activities [95,98,99]. 

Finally, concerning the hypothesis of inferential type (Hypothesis 3), we showed that self-regulation level in adolescents positively carry with him the self-efficacy, procrastination (inversely), and achievement levels. Consequently, procrastination is consolidated as a dys-regulatory variable [100,101,102].

On the other hand, as pointed out by previous studies and the present study results, parental support influences the academic achievement of students and functions as a protective factor at the contextual level [64,103,104]; a key factor is the mother’s role and her academic level as it contributes significantly to the academic achievement of the children [105]. Thus, although Colombian adolescents have cognitive, behavioral, and personal self-regulation tools for their learning processes, if their social context does not promote academic goal achievement, it is more probable that de-regulatory behaviors appear in their educational context [52].

Complementarily, the results show that self-efficacy is positively associated with self-regulation (Vincent et al., 2021) [106], but self-regulation has a stronger predictive force for academic achievement. These results are consistent with the idea that self-efficacy is a previous variable of the initial self-regulation phase, that is, only a part of it (Zimmerman and Labuhn, 2012) [107].

### 4.2. Limitations

The present investigation also has limitations that need to be analyzed. Among them is the specificity of the sample; for further investigation, it would be interesting to include a broader demographic (inclusion of more public and private schools) for better analysis depth. In addition, because the empirical evidence of studies that analyze the variables used in the present study and that include high school Latin American students is scarce, it is necessary to widen the number of studies that focus on analyzing how self-efficacy and self-regulation contribute to academic achievement. Likewise, other instruments could be employed that help to evaluate the effect that context has (not only the parent’s educational level) on academic achievement, regulation, procrastination, and well-being of adolescents.

## 5. Conclusions

The results showed that self-regulation is the most potent and predictive personal variable of procrastination and achievement, being positively associated with self-efficacy. Complementarily, the parents’ academic level is also predictive, however, to a lesser extent.

Being a woman and belonging to the higher age group negatively predicted achievement. In Colombia, according to a study by the National Statistical Administrative Department (DANE for its acronym in Spanish) (2020) [108], the gender gap begins in early ages because girls dedicate more time to domestic labors, to non-remunerated care, and the economic support of their families, distancing them from academic contexts. For this reason, the obtained results belonging to the groups mentioned above are a risk factor of academic achievement during adolescence in Colombia.

However, with the implementation of strategies of academic counseling in which the adolescent girls are empowered to generate confidence in their academic skills, this matter could be minimized, being academic confidence a predictor of exemplary academic achievement [109].

On the other hand, because home is a strong predictor of academic achievement, especially when living in regulatory environments that promote psychological well-being [109,110], it raises the need to involve parents in the educational activities and offer them upbringing tools that allow them to support, encourage and direct education and emotional regulation of their children.

## Figures and Tables

**Figure 1 ijerph-18-08944-f001:**
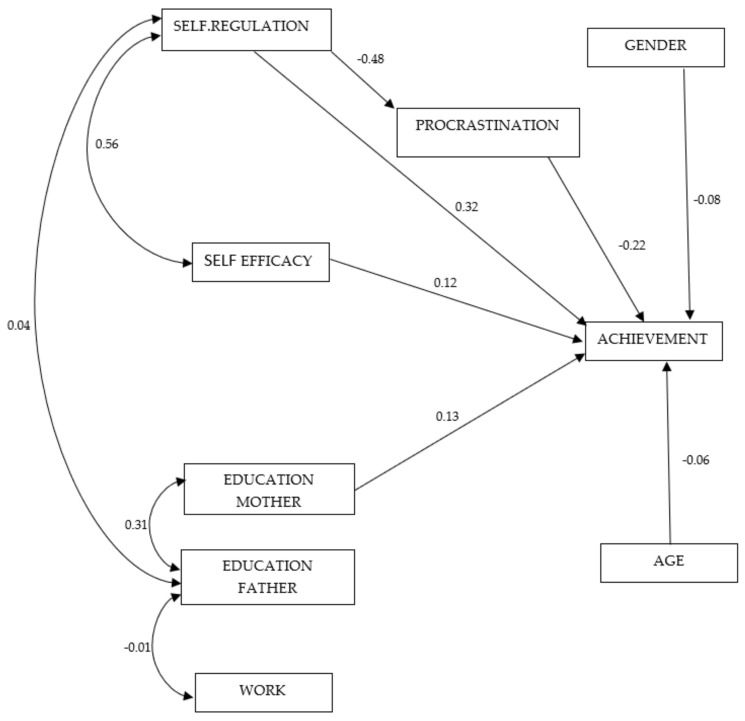
Obtained structural model prediction.

**Figure 2 ijerph-18-08944-f002:**
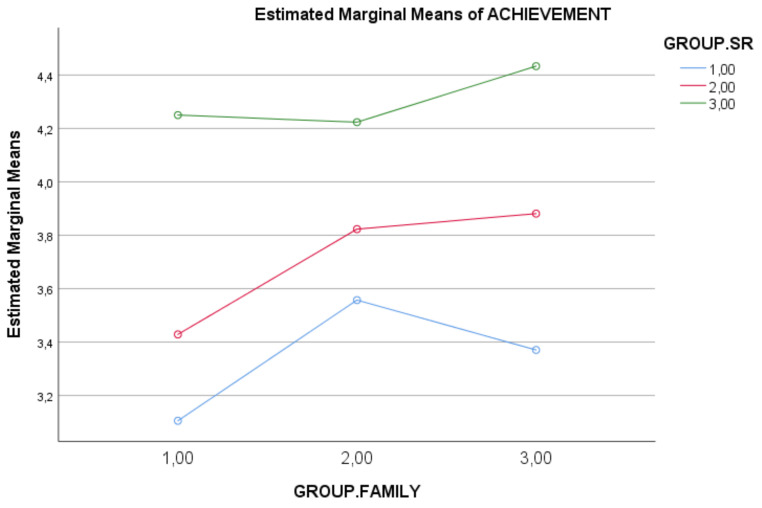
Self-regulation students level (GROUP.SR; 1.00 = Low; 2.00 = Medium; 3.00 = High) and socio-educative family level (GROUP.FAMILY; 1.00 = Low; 2.00 = Medium; 3.00 = High) effects on the adolescent’s academic achievement.

**Table 1 ijerph-18-08944-t001:** Descriptive and normative values of the sample (*n* = 430).

Statistical Index	Age	Gender	Self−Regulation	Procrastination	Self−Efficacy	Education Mother	Education Father	Work Type	Academic Achievement
Range	(11–18)	(1–2)	(1–5)	(1–5)	(1–4)	(1–6)	(1–6)	(1–15)	(1–10)
Minimum	11	1	1.99	1.00	1.11	1.00	1.00	1	1
Maximum	18	2	5.42	5.00	4.00	6.00	6.00	15	10
Mean	14.50	1.45	3.79	2.81	2.9664	4.22	4.04	6.61	3.84
Standard Deviation	1.93	0.498	0.57	0.81	0.59642	1.10	1.25	4.075	0.764
Asymmetry	−0.018	0.206	−0.09	−0.251	−0.144	−0.529	−0.443	0.295	−0.502
Standard asymmetry error	0.118	0.118	0.11	−0.118	0.118	0.121	0.127	0.118	0.118
Kurtosis	−1.048	−1.967	0.031	−0.475	−0.579	0.420	0.099	−0.942	0.661
Standard Kurtosis error	0.235	0.235	0.235	0.235	0.232	0.241	0.253	0.235	0.235

**Table 2 ijerph-18-08944-t002:** Bivariate correlations (*n* = 430).

Variable	1	2	3	4	5	6	7	8
1. Age 2. Gender 3.Self-Regulation								
4. Procrastination			−0.484 ***					
5.Self-Efficacy			0.559 ***	−0.318 **				
6.Education Mother					0.168 ***			
7.Edu.Father			0.107 *	−0.108 *	0.166 ***	0.40 ***		
8.Work Type								
9.Academic Achievement	−0.022	−0.111 *	0.494 ***	−0.427 ***	0.377 ***	0.201 ***	0.191 ***	

Note. * *p* < 0.05, ** *p* < 0.01, *** *p* < 0.001.

**Table 3 ijerph-18-08944-t003:** Models of structural linear results of the variables.

Model	Chi^2^	DF	CH/df	SRMR	*p* <	NFI	RFI	IFI	TLI	CFI	RMSEA	HOELT *p* < 0.05	HOELT *p* < 0.1
1 (4 Factors)	67.594	2(14–12)	33.797	0.1185	0.001	0.834	0.530	0.847	0.537	0.846	0.276	39	59
2 (7 Factors)	53.177	14(35–21)	3.798	0.0567	0.001	0.839	0.840	0.919	0.877	0.918	0.081	192	236
3 (9 Factors)	34.203	18(44–26)	1.90	0.0518	0.01	0.934	0.898	0.968	0.947	0.967	0.046	363	473

Note. *Model 1*: self-regulation and self-efficacy → procrastination → achievement *Model 2*: self-regulation → procrastination → achievement; self-regulation and self-efficacy → achievement; mother education and father education and work → achievement; *Model 3*: gender and year → achievement; SR → procrastination → achievement; SR and self-efficacy → procrastination → achievement; mother education and father education and work → achievement; DF = degrees of freedom; CH/df = Chi-square/degrees of freedom; SRMR = standardized root—mean-square residual; *p* ≤ probability level; NFI = normed fit index; RFI = relative fit index; IFI = incremental fit index; TLI = Tucker–Lewis index; CFI = comparative fit index; RMSEA = root mean square error of approximation; HOELT = Hoelter index.

**Table 4 ijerph-18-08944-t004:** Total, indirect, and direct effects of the variables in this study, and 95% bootstrap confidence intervals (CI). Bootstrapping sample size = 430.

Predictor Variable	Criterion Variable	Total Effect	CI (95%)	Direct Effect	CI (95%)	Indirect Effect	CI (95%)	Results, Effects	CI (95%)
Gender →	Achievement	−0.08	[−0.05, −0.09]	−0.07	[−0.05, −0.09]	0.00	[−0.02, 0.02]	Direct only	[−0.05, −0.09]
Year group →	Achievement	−0.08	[−0.06, −0.10]	−0.08	[−0.06, −0.10]	0.00	[−0.02, 0.02]	Direct only	[−0.06, −0.10]
Self-Regulat →	Procrastination	−0.48	[−0.47, −0.50]	−0.48	[−0.47, −0.50]	0.00	[−0.02, 0.02]	Direct only	[−0.47, −0.50]
Self-Regulat →	Achievement	0.32	[0.30, 0.34]	0.32	[0.30, 0.34]	0.110	[0.09, 0.13]	Partial mediation	[0.09, 0.13]
Self-Efficacy →	Achievement	0.123	[0.09, 0.15]	−0.19	[−0.15, −0,21]	0.00	[−0.02, 0.02]	Direct only	[0.09, 0.15]
Gender →	Achievement	−0.07	[−0.05, −0.09]	−0.07	[−0.05, −0.09]	0.00	[−0.02, 0.02]	Direct only	[−0.05, −0.09]
Education Mother →	Achievement	0.13	[0.11, 0.15]	0.00	[−0.02, 0.02]	0.00	[−0.02, 0.02]	Direct only	[0.11, 0.15]
Education Father →	Achievement	0.31	[0.29, 0.33]	0.00	[−0.02, 0.02]	0.31	[0.29, 0.33]	Indirect only	[0.29, 0.33]
Work →	Achievement	−0.01	[−0.03, 0.01]	0.00	[−0.03, 0.04]	−0.01	[−0.03, 0.01]	Indirect only	[−0.03, 0.01]

**Table 5 ijerph-18-08944-t005:** Self-regulation (SR) and family education level (FE) effects on the adolescent’s academic achievement.

Variable										*F* (Pillai)	*p* <	*r* ^2^	Post-Hoc
SR level (SR)	Low	Low	Low	Medium	Medium	Medium	High	High	High	*F*(6840) = 23,902	0.0001	0.146	
SEC level	Low (*n* = 19)	Medium (*n* = 70)	High (*n* = 27)	Low (*n* = 116)	Medium (*n* = 28)	High (*n* = 113)	Low (*n* = 16)	Medium (*n* = 85)	High (*n* = 30)	*F*(6480) = 2169	0.05	0.015	
Procrastination	3.14 (0.57)	3.26 (0.65)	3.27 (0.80)	3.01 (0.68)	2.81 (0.68)	2.79 (0.69)	2.32 (0.77)	2.48 (0.94)	2.11 (0.74)	SR,*F*(2430) = 32,682	0.0001	0.134	SR,3>2>1
Self-Efficacy	2.51 (0.50)	2.56 (0.51)	2.66 (0.60)	2.80 (0.58)	2.94 (0.21)	2.94 (0.53)	3.24 (0.63)	3.26 (0.54)	3.44 (0.43)	SR,*F*(2430) = 47,733	0.0001	0.185	SR,3>2>1
Achievement	3.11 (0.56)	3.56 (0.79)	3.37 (0.92)	3.43 (0.79)	3.46 (0.67)	3.88 (0.63)	4.25 (0.56)	4.22 (0.79)	4.43 (0.92)	SR,*F*(2430) = 42,325 FE,*F*(2430) = 4322	0.0001 0.01	0.167 0.020	SR,3>2>1 FE,3,2>1

Note. SEC level (socio-educational context level) is the average of the educational level of the mother and the father (range: 1–6 points) and of work type (range: 1–15 points).

## Data Availability

The datasets used and/or analyzed during the current study are available from the corresponding author on reasonable request.

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
