# Peer review of "Effect of Personal and Contextual Factors of Regulation on Academic Achievement during Adolescence: The Role of Gender and Age"

_ijerph, 2021, doi:10.3390/ijerph18178944_

Round 1

Reviewer 1 Report

In this study, the authors intended to predict the academic achievement of Colombian adolescents by multiple hypothetically related variables such as self-regulation, procrastination, academic self-efficacy, and family background. From online surveys consist of self-report scales administered to 430 students between the ages of 11 and 18 were derived conclusions that both self-regulation, self-efficacy, procrastination, gender, and maternal education affected students’ academic achievement. Also, the effect of self-regulation on academic achievement was partially mediated by self-reported procrastination. The article may have some merit to the current literature, especially reporting data on not commonly addressed populations (location and age). I also commend authors for profound preliminary data analysis and controlling for multiple variables. However, the contributions of the study are limited by the lack of coherent theoretical background, unacknowledged methodological limitations, lack of reporting, and inappropriate causal inferences. Tables and figures do not adhere to the APA style and add unnecessary confusion to the data report. I suggest major revisions including conceptuality clarity, more details about the outcome measures, data collection procedures, and analytical decisions.

First of all, the study is crucially limited by the lack of a theoretical framework that would clearly define the considered variables and outline the potential hypothesized association among them. For instance, there is a clear distinction between self-regulation and self-regulated learning (e.g. Schunk, 2008; Alexander, 2008; Lajoie, 2008) that is not acknowledged. Such delineation affects methodological decisions. This is demonstrated by the disconnect; the instrument used in the study assesses self-regulation whereas the introduction refers to references to evidence and theories that are based on self-regulated learning concepts. Such mismatch must be resolved. Furthermore, the utilized measure of self-regulation (Garzón-Umerenkova et al., 2017) has not been validated on the adolescent population, but merely on the population 17+. The authors have to provide further evidence of the instrument validity of the assessed population. The lack of conceptual clarity applies to other variables used in the study. For example, there is no description of the measurement of academic achievement that is the main outcome variable.

The description of the data collection procedure must be explained more in detail: How we the school recruited? How was anonymity and confidentiality of participants warranted? Who administered the online forms to students? What instructions were provided to students? Where did students take the online survey? Was there any incentive or other motivation for students to complete the survey?

The lack of clarity in language must be dispelled. Hypotheses (p.4, 177 & 184) need to be described more precisely and the hypothesized predictions have to be justified by the literature. For instance, the choice of wording “associated and predict“ is unfortunate as the authors imply that they would be testing both correlational and causal relationships, however, the study design is clearly correlational and causation cannot be considered. Hypothesis 3 in section 1.5.2 draws on further potential difficulties (section 3.4) described in the paragraph below. Similar tendencies of language suggesting causal inferences are suggested in conclusions.

The tested models and their adoption should be better communicated. The authors would drastically improve the understanding of the data by drawing multiple figures of the hypothesized models that were tested or other descriptions of the analytical decisions they made before they arrived at their final model. One charted SEM model does not give sufficient insight into the decisions the authors made about their analysis. Furthermore, the AMOS screenshot of their tested SEM model is not appropriate for scientific publication. Authors have to follow APA formatting in presenting all their tables and charts and provide sufficient explanation on how they arrived at their models that either stems from theoretical discussion in the introduction or previous analytical conclusions. Furthermore, I could not find the reason why the authors decided to include further section 3.4 that includes analyses of variances of specific variables. The section introduces uninvited questions, such as dividing variables of self-regulation and academic context? (this is a new variable) into arbitrary categories that do not have any substantial reasoning. The section also introduces the confusion about causality that does not have a place in the presented study.

Author Response

General

In this study, the authors intended to predict the academic achievement of Colombian adolescents by multiple hypothetically related variables such as self-regulation, procrastination, academic self-efficacy, and family background. From online surveys consist of self-report scales administered to 430 students between the ages of 11 and 18 were derived conclusions that both self-regulation, self-efficacy, procrastination, gender, and maternal education affected students’ academic achievement. Also, the effect of self-regulation on academic achievement was partially mediated by self-reported procrastination. The article may have some merit to the current literature, especially reporting data on not commonly addressed populations (location and age). I also commend authors for profound preliminary data analysis and controlling for multiple variables. However, the contributions of the study are limited by the lack of coherent theoretical background, unacknowledged methodological limitations, lack of reporting, and inappropriate causal inferences. Tables and figures do not adhere to the APA style and add unnecessary confusion to the data report. I suggest major revisions including conceptuality clarity, more details about the outcome measures, data collection procedures, and analytical decisions.

Introduction

  • First of all, the study is crucially limited by the lack of a theoretical framework that would clearly define the considered variables and outline the potential hypothesized association among them. For instance, there is a clear distinction between self-regulation and self-regulated learning (e.g. Schunk, 2008; Alexander, 2008; Lajoie, 2008) that is not acknowledged. Such delineation affects methodological decisions. This is demonstrated by the disconnect;

Answer: the relationship in SR and between SRL in the text has been reviewed and clarified: Lines 78-82, 84-100, 124-130

  • The lack of clarity in language must be dispelled. Hypotheses (p.4, 177 & 184) need to be described more precisely and the hypothesized predictions have to be justified by the literature. For instance, the choice of wording “associated and predict“ is unfortunate as the authors imply that they would be testing both correlational and causal relationships, however, the study design is clearly correlational and causation cannot be considered.

Answer: the hypotheses have been revised and better substantiated. It was made clear that the association and structural prediction hypotheses are linear and not causal.

  • Hypothesis 3 in section 1.5.2 draws on further potential difficulties (section 3.4) described in the paragraph below.

Answer: the text has been revised and redefined. Please, lines 292-297

  • Similar tendencies of language suggesting causal inferences are suggested in conclusions.

Answer: the only inferences, potentially causal, are those derived from the third hypothesis and the ANOVAs and MANOVAs. The text of the discussion has been revised.

Method

  • The instrument used in the study assesses self-regulation whereas the introduction refers to references to evidence and theories that are based on self-regulated learning concepts. Such mismatch must be resolved. Furthermore, the utilized measure of self-regulation (Garzón-Umerenkova et al., 2017) has not been validated on the adolescent population, but merely on the population 17+. The authors have to provide further evidence of the instrument validity of the assessed population. The lack of conceptual clarity applies to other variables used in the study. For example, there is no description of the measurement of academic achievement that is the main outcome variable.

Answer: The requested values have been inserted.

  • The description of the data collection procedure must be explained more in detail: How we the school recruited? How was anonymity and confidentiality of participants warranted? Who administered the online forms to students? What instructions were provided to students? Where did students take the online survey? Was there any incentive or other motivation for students to complete the survey?

Answer: An operational definition of this variable has been inserted.

Results

  • Results. The tested models and their adoption should be better communicated. The authors would drastically improve the understanding of the data by drawing multiple figures of the hypothesized models that were tested or other descriptions of the analytical decisions they made before they arrived at their final model. One charted SEM model does not give sufficient insight into the decisions the authors made about their analysis. Furthermore, the AMOS screenshot of their tested SEM model is not appropriate for scientific publication. Authors have to follow APA formatting in presenting all their tables and charts and provide sufficient explanation on how they arrived at their models that either stems from theoretical discussion in the introduction or previous analytical conclusions.

Answer:  The tested models have been specifically explained; see lines 329-335.  The place of the tables in the text has been changed, for a better understanding.  Regarding figures and tables, the formats allowed by this Journal have been adopted.

  • Furthermore, I could not find the reason why the authors decided to include further section 3.4 that includes analyses of variances of specific variables. The section introduces uninvited questions, such as dividing variables of self-regulation and academic context? (this is a new variable) into arbitrary categories that do not have any substantial reasoning. The section also introduces the confusion about causality that does not have a place in the presented study.

Answer: The formation of this compound variable (sum of the level of studies of the father, mother and work), not previously analyzed, has been explained in the analysis of data (lines 292-297) and results (lines 360 and 368).

Reviewer 2 Report

A relevant and significant problem is analysed. The publication has been
prepared properly, presenting the results of the conducted research. Here are a few observations which, in my opinion, would allow a clearer understanding of the content of the article and improve its quality:

- I would recommend specifying the title of the article; I understand that you want to “fit in” everything in it, but methodologically it seems very long.

- It is not clear to me how “parents’ academic level” was assessed, in some places “the socio-academic level” is also mentioned – is it the same? Maybe parents’ education was meant. It is necessary to explain how their levels are assessed, whether the results of both parents are assessed or of one parent and the like, if the research participant is from a one-parent family. The conclusions also refer to the family in general but are based on the results where the mother’s role is distinguished as more important.

- In some pictures, the original language is left, so translation into English is required (Figure 2).

- The conclusions could be made more specific, now they look more like the discussion that precedes them.

Considerable work has been done, congratulations to the authors and I wish good luck.

Author Response

1)TITLE: I would recommend specifying the title of the article; I understand that you want to “fit in” everything in it, but methodologically it seems very long.

Answer: Thank you. It has been simplified with a simpler and more inclusive title.

2) It is not clear to me how “parents’ academic level” was assessed, in some places “the socio-academic level” is also mentioned – is it the same? Maybe parents’ education was meant. It is necessary to explain how their levels are assessed, whether the results of both parents are assessed or of one parent and the like, if the research participant is from a one-parent family. The conclusions also refer to the family in general but are based on the results where the mother’s role is distinguished as more important.

Answer: The socio-educational level of the parents was evaluated through (1) a self-report of the level of studies (educational level), together with (2) the type of work (socio-economic level). The current label has been replaced by socio-educational level that encompasses both variables. The type of family, single parent vs biparental, was not considered.

3) In some pictures, the original language is left, so translation into English is required (Figure 2).

Answer. Thanks. They have been revised and translated into English.

4) The conclusions could be made more specific, now they look more like the discussion that precedes them.

Answer: They have been revised and adapted.

5) Considerable work has been done, congratulations to the authors and I wish good luck.

Answer: Thanks for the contributions

Reviewer 3 Report

This article was difficult to "unpack".  The English writing was not standard. (See Line 14:    . . and informed their academic achievement.  I'm not sure what this means or how it occurred).  There are many instances where the unusual sentence structure or non-standard word usage blurred clarity and meaning.  Articles needs an editor!

The authors did not address their operational definition of academic achievement which is the central dependent variable.  Article implies (Line 229) that achievement data were students' self report and had a 1-10 as suggested by a Table 1 heading.  What? Why? Rationale? This might be a major validity threat, despite a lot of stat number crunching.

I have no quibbles with overall statistical approach. There are a jumble of tables, variable acronyms, and notes related to Tables 3, 4, and 5.  All of this was very difficult to process.  Needs simplification and clarity.  For instance, use complete labels in Figure 1 for clarity.

Article does not attempt to explain why self-regulation predicts and self-efficacy does not predict.  Article's abstract calls for intervention, but suggested interventions are not offered.  Thus, the reader is left with "So What?"

Author Response

General

  • This article was difficult to "unpack".  The English writing was not standard. (See Line 14:    . . and informed their academic achievement.  I'm not sure what this means or how it occurred).  There are many instances where the unusual sentence structure or non-standard word usage blurred clarity and meaning.  Articles needs an editor!

Answer: It has been revised again by a foreign translator

Method

  • The authors did not address their operational definition of academic achievement which is the central dependent variable.  Article implies (Line 229) that achievement data were students' self report and had a 1-10 as suggested by a Table 1 heading.  What? Why? Rationale? This might be a major validity threat, despite a lot of stat number crunching.

Answer: Students reported their average grade for the previous year. Each year the average grade of the student is calculated, from all the averages obtained in all the subjects, to put in their academic record. Although this data was obtained in a self-reported way by each student, it has a high reliability, since it is not calculated by the student nor is it an appreciation. It is an objective data and a very exact correlate of the average of the performance (range: 1-10 points). See line 229-233.

  • I have no quibbles with overall statistical approach. There are a jumble of tables, variable acronyms, and notes related to Tables 3, 4, and 5.  All of this was very difficult to process.  Needs simplification and clarity.  For instance, use complete labels in Figure 1 for clarity.

Answer: The acronyms of the Tables and Figures have been simplified. They have all been in English. See, line 312

Discussion

  • Article does not attempt to explain why self-regulation predicts and self-efficacy does not predict.  

Answer. Thanks. An explanation has been inserted in the text. See 449-452.

  • Article's abstract calls for intervention, but suggested interventions are not offered.  Thus, the reader is left with "So What?"

Answer. General intervention strategies are exposed. See lines 464-477.

Round 2

Reviewer 1 Report

The authors made significant revisions in their manuscript, however, they did not address all points sufficiently to remove the major flaws described in the previous review. I will clarify

In case of the conceptual clarity, the authors very well described the concept of academic achievement and its measurement. However, their distinction between self-regulation (SR) and self-regulated learning (SRL) is still insufficient. For instance, the edited descriptions of SR and SRL between lines 78 and 88 do not contrast the two concepts but merely define them in isolation. However, there is a crucial need to explain how these two terms differ from each other and why did they choose to measure SR instead of SRL. Without a clear distinction between the two concepts, the authors cannot use SRL theory in their introduction as the findings from SRL research do not simply apply to the conceptual findings they focus on. Another sensible option would be dropping the mention of the SRL from the introduction and focusing solely on SR.

I take an issue with implying that the current study could (even potentially) suggest any causation. The current study design does not simply enable researchers to make causal implications (not even the hypothesis 3). The language in the description of hypotheses is still confusing. Words `associate` and `predict` should not be used in the same sentence. The word ‘determine’ also bears a connotation of causality. Causal inferences could be merely discussed as a suggestion for future research, however, the current data cannot be interpreted this way, unless making a substantial change to the research design and collecting new data.

The reports of the psychometric properties of the utilized scales appear inconsistent. Authors provide reliability and factor analytical indices, or Rasch model results for some variables, but not for other variables. No explanation for why some variables were treated by selected analytical approaches unlike other variables was provided. Authors should be more consistent in their reporting and/or provide reasoning for the statistical decisions they made.

The description of the procedures was sufficient.

I do not consider the image of the SEM pathways in the form of an AMOS output screenshot as acceptable.

Author Response

The authors made significant revisions in their manuscript, however, they did not address all points sufficiently to remove the major flaws described in the previous review. I will clarify:

  • Self-Regulation vs Self-Regulated learning. In case of the conceptual clarity, the authors very well described the concept of academic achievement and its measurement. However, their distinction between self-regulation (SR) and self-regulated learning (SRL) is still insufficient. For instance, the edited descriptions of SR and SRL between lines 78 and 88 do not contrast the two concepts but merely define them in isolation. However, there is a crucial need to explain how these two terms differ from each other and why did they choose to measure SR instead of SRL. Without a clear distinction between the two concepts, the authors cannot use SRL theory in their introduction as the findings from SRL research do not simply apply to the conceptual findings they focus on. Another sensible option would be dropping the mention of the SRL from the introduction and focusing solely on SR.

Answer: Adjustments have been made between lines 87-100. More explanation and its relationship with self-regulatory learning have been incorporated. It is important for the reader to understand that SR (behavioral and general  personal presage variable) is a factor that predicts the SRL process variable, in learning situations. Therefore, this variable has been used (SR)  and self-regulated learning (SRL) has not been measured.

  • I take an issue with implying that the current study could (even potentially) suggest any causation. The current study design does not simply enable researchers to make causal implications (not even the hypothesis 3). The language in the description of hypotheses is still confusing. Words `associate` and `predict` should not be used in the same sentence. The word ‘determine’ also bears a connotation of causality. Causal inferences could be merely discussed as a suggestion for future research, however, the current data cannot be interpreted this way, unless making a substantial change to the research design and collecting new data.

Answer: The third hypothesis has been adjusted; The word "determine" has been changed in the text. Lines: 181, 316, 381, 383. Thanks.

3) Instruments The reports of the psychometric properties of the utilized scales appear inconsistent. Authors provide reliability and factor analytical indices, or Rasch model results for some variables, but not for other variables. No explanation for why some variables were treated by selected analytical approaches unlike other variables was provided. Authors should be more consistent in their reporting and/or provide reasoning for the statistical decisions they made.

Answer:  The descriptions of the instruments and their psychometric indicators are those reported by previous validation studies. Different articles are cited for each instrument and the results correspond to what the related authors carried out in the cited studies. However, the information from the PASS test validation study has been completed. See, line 220-226.

5) Procedure. The description of the procedures was sufficient.

Answer. Thanks

6) Image. I do not consider the image of the SEM pathways in the form of an AMOS output screenshot as acceptable.

Answer: The image has been changed. See line 298

Reviewer 3 Report

Study is improved, although I am still concerned about the assumption that self-reported grades reflect achievement.  I will accept that self-reported grades might correlate with study's socio variables (e.g., procrastination).  So I side with authors.

Tables are improved.

Author Response

1)Study is improved, although I am still concerned about the assumption that self-reported grades reflect achievement.  I will accept that self-reported grades might correlate with study's socio variables (e.g., procrastination).  So I side with authors.

Answer: Thank you.